# HST-bench: A Cognitive-Science Grounded Benchmark for Hierarchical Spatial Thinking in Large Language Models

## Abstract

Large language models (LLMs) show strong potential for real-world applications, yet their deployment in domains requiring deep interaction with the physical world hinges on robust spatial ability. Existing evaluations are constrained by a flawed, *task-driven* paradigm that probes surface-level perception, lacking the cognitive depth and theoretical guidance needed for true diagnostic precision. To address this, we introduce **HST-bench**, a benchmark for **H**ierarchical **S**patial **T**hinking that instigates a paradigm shift to *theory-driven* evaluation. Grounded in the National Research Council's theory, HST-bench organizes assessment along three core cognitive dimensions: Representational Perception, Representational Transformation, and Spatial Reasoning. Spanning 1,629 problems across 10 sub-dimensions, our tasks require dynamic operations such as coordinate transformation and symmetry, demanding deep spatial representation and reasoning. Comprehensive evaluations reveal that a **"thinking mechanism"** is critical for advanced spatial tasks. We further observe a strong positive correlation between general and spatial capabilities, and importantly, limited gains from multimodal inputs, highlighting the current primacy of reasoning over perception. HST-bench offers a principled, cognitively grounded path toward diagnosing and advancing the spatial intelligence of large models.

## 1 Introduction

Large language models (LLMs), with their outstanding cognitive and reasoning capabilities, have demonstrated significant application potential in key areas such as traffic optimization (Liu et al., 2024a), robot control (Vemprala et al., 2023; Ma et al., 2024; Song et al., 2025), urban planning (Zhu et al., 2024), and autonomous driving (Yang et al., 2024b; Zhang et al., 2025). However, whether these models can truly handle tasks that require deep interaction with the physical world hinges on a core capability that has not yet been fully deconstructed—**spatial ability**. A key reason for this gap is that current evaluations are constrained by a flawed paradigm: most benchmarks are predominantly *task-driven* rather than *theory-driven*, which limits diagnostic precision of a model's true capabilities, comparability across studies, and guidance for model improvement. To make this distinction concrete, task-driven benchmarks typically assemble datasets and metrics around particular applications or question formats, whereas theory-driven benchmarks start from principled models of spatial cognition to derive measurable abilities and evaluation criteria. Although several studies have proposed spatial evaluations for LLMs—such as SpatialVLM (Chen et al., 2024) for 3D VQA, CA-VQA (Daxberger et al., 2025) for indoor scene understanding, and Open3DVQA (Zhan et al., 2025) for absolute spatial relationships—current evaluation paradigms exhibit three fundamental limitations.

**First**, current evaluation tasks lack cognitive depth. Existing benchmarks largely focus on static spatial relationship identification (e.g., "Is A to the left of B?") or absolute metrics (e.g., "What is the distance between A and B?"), which primarily assess a model's surface-level perception but overlook higher-order operations essential for comprehensive spatial intelligence, such as mental rotation, coordinate system transformation, or multi-step trajectory prediction. **Second**, current benchmarks are predominantly task-driven. Such evaluations encourage models to exploit dataset-specific heuristics, achieving superficial task success without demonstrating genuine competence. They also

hinder comparability across studies and fail to provide fine-grained diagnostic signals that can guide principled model improvements. **Third**, existing frameworks lack systematic guidance from cognitive science. Without such theoretically grounded dimensions, benchmarks remain fragmented and cannot reflect the hierarchical structure of spatial thinking established in psychology and education research. This limits their ability to both diagnose the shortcomings of current models and meaningfully guide their future development.

To address these deficiencies, we introduce **HST-bench**, a novel benchmark for evaluating the spatial abilities of LLMs, grounded in established cognitive science. Fundamentally departing from prior work, HST-bench instigates a paradigm shift from being *task-driven* to *theory-driven*. As illustrated in the lower part of Figure 1, we operationalize the hierarchical theory of spatial thinking proposed by the U.S. National Research Council et al. (2005) into a framework that requires models to complete complex spatial operations such as coordinate transformation, rotation, and symmetry, rather than merely identifying static object positions. This framework systematically assesses models across three core cognitive dimensions: **1) Representational Perception**, the ability to understand and reconstruct basic spatial information; **2) Representational Transformation**, the ability to perform dynamic mental operations such as rotation, symmetry, and projection; and **3) Spatial Reasoning**, the ability to execute multi-step, long-chain logical inference in complex spatial scenarios. These dimensions not only align with how humans develop spatial intelligence, but also map naturally to an LLM's processing pipeline—from encoding spatial structure (representation), to manipulating it (transformation), to chaining inferences over it (reasoning).

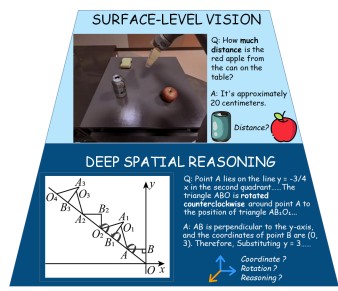

Figure 1: Surface-level perception vs. deep spatial reasoning. While existing benchmarks test basic visual perception, our approach requires complex, multi-step inference.

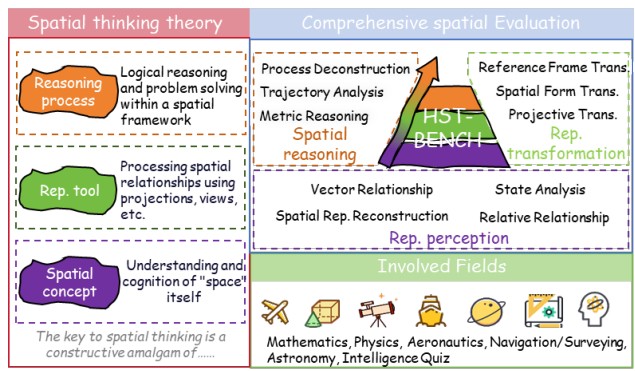

Figure 2: **Overview of HST-bench.** Based on spatially thinking theory, HST-bench maps the three core elements of spatial cognition to three distinct dimensions of cognitive abilities.

Our main contributions are summarized as follows:

- We introduce **HST-bench**, the first evaluation benchmark for the spatial ability of LLMs that instigates a paradigm shift to be systematically grounded in and driven by cognitive science. It assesses three core cognitive dimensions, moves beyond surface-level perception to enable a deep, diagnostic evaluation of a model's spatial intelligence.

- We demonstrate that HST-bench presents a significant challenge for contemporary LLMs, especially in complex spatial transformation and reasoning. Our comprehensive evaluations reveal a substantial performance gap between models and human experts.

- Through comprehensive evaluations, we identify that a **"thinking mechanism"** is the critical factor for advanced spatial tasks, suggesting that future advancements in spatial intelligence will depend more on improving structured reasoning architectures than on merely scaling model size.

## 2 RELATED WORK

**Spatial Ability in Cognitive Science.** The study of spatial ability is deeply rooted in cognitive science, which provides a theoretical gold standard for its assessment. Foundational theories, such as

Carroll's three-stratum model, identified a distinct cognitive layer for "forming mental representations to solve spatial problems" (Carroll, 1993). The National Research Council further synthesized this field by deconstructing spatial thinking into three core components: understanding spatial concepts, utilizing representational tools, and executing reasoning processes (Council et al., 2005). This established framework posits that human spatial intelligence is not a monolithic capability but a hierarchical system involving perception, mental transformation, and logical inference. Unlike the task-driven evaluations prevalent in LLMs, these cognitive theories emphasize the multi-layered structure of spatial thought. Our work, HST-bench, is the first to systematically operationalize this hierarchical cognitive framework for evaluating LLMs.

**Benchmarking Spatial Abilities in LLMs.** Recent research has begun to probe the spatial capabilities of LLMs, primarily through vision-centric benchmarks. For instance, SpatialVLM (Chen et al., 2024) introduced metric spatial reasoning from real-world images, and CA-VQA (Daxberger et al., 2025) focused on parsing object topological structures in 3D scenes. In video, VSI-Bench (Yang et al., 2024a) established the first benchmark for visual-spatial intelligence. While these works are foundational, they often conflate visual understanding with spatial reasoning. As noted by benchmarks like Open3DVQA (Zhan et al., 2025), current models struggle with complex spatial tasks. A significant gap remains: existing evaluations lack systematic coverage of the multi-layered cognitive structures of spatial intelligence identified by cognitive science, which our work aims to address.

**Paradigms for Domain-Specific Evaluation.** The methodology of evaluating domain-specific capabilities in LLMs offers a valuable paradigm for our research. Foundational benchmarks like MMLU (Hendrycks et al., 2021) and GSM8K (Cobbe et al., 2021) established robust methods for assessing knowledge across diverse subjects. More advanced frameworks, such as MathBench (Liu et al., 2024b) and MathVista (Lu et al., 2024), introduced hierarchical structures to test the transfer from theoretical understanding to practical application. These benchmarks inspire our approach. By adapting their principles of hierarchical and multifaceted assessment, HST-bench is the first to apply this rigorous evaluation methodology to the core facets of spatial thinking, thereby providing a more comprehensive and theoretically-grounded measure of an LLM's spatial intelligence.

## 3 METHODOLOGY

The Methodology section will revolve around three main modules: spatial thinking theory, the design of the evaluation framework, and the construction of the dataset.

### 3.1 SPATIAL THINKING THEORY

We ground our work in the spatial thinking theory proposed by the U.S. National Research (Council et al., 2005), which characterizes spatial cognition through three interrelated elements: *concepts of space*, *tools of representation*, and *processes of reasoning*. Together, these elements describe how humans perceive, represent, and reason about spatial information, and they provide the theoretical foundation for our evaluation framework.

**Concepts of Space**: the cognitive foundation of spatial thinking, covering the understanding of spatial attributes, spatial relationships, and semantic structures that allow agents to construct problem domains. **Tools of Representation**: the external and internal representational systems (e.g., diagrams, symbolic descriptions, coordinate systems) that concretize spatial concepts and enable structured problem solving. **Processes of Reasoning**: the dynamic operations applied to representations, from perceiving static object properties to performing mental transformations (e.g., rotation, projection) and drawing higher-level inferences. This tripartite structure not only explains how spatial intelligence develops in humans but also maps naturally onto the evaluation of LLMs: assessing their ability to encode spatial concepts, manipulate representations, and integrate them for reasoning. It serves as the conceptual backbone for the three evaluation dimensions introduced in Section 3.2.

### 3.2 EVALUATION FRAMEWORK DESIGN

Building on the National Research Council's three-element theory of spatial thinking (concepts of space, tools of representation, processes of reasoning), we design an evaluation framework that operationalizes these principles into three core dimensions: Representational Perception, Represen-

tational Transformation, and Spatial Reasoning. Unlike task-driven benchmarks that emphasize isolated outputs, this framework decomposes spatial intelligence into cognitive components, enabling systematic and fine-grained diagnosis.

**Representational Perception** (corresponding to concepts of space) evaluates the ability to encode and reconstruct spatial information, covering both static and dynamic attributes. Subtasks include spatial scene reconstruction, vector relations, relative relations, and state analysis. **Representational Transformation** (tools of representation) assesses the ability to mentally manipulate spatial structures under changes in perspective, scale, or form. Subtasks include reference-frame transformation, spatial-form transformation (e.g., rotation, symmetry), and projective transformation. **Spatial Reasoning** (processes of reasoning) examines the integration of perception and transformation for multi-step inference, such as process deconstruction, trajectory analysis, and metric reasoning in complex scenarios. The detailed conceptions are provided in the Appendix A.2 and Table5.

### 3.3 DATASET CURATION

**Data Sources and Collection**   To systematically assess diverse spatial thinking abilities, we collected problems from mathematics, physics, navigation, surveying, and intelligence testing, guided by our dimensional framework. **Annotation Process**: two master's students in science and engineering conducted question collection, image and formula processing, dimensional labeling, and data cleaning using a web-based tool. Each question was labeled with dimension and answer type, then standardized into a multiple-choice format; for image-based questions, all visual information was made fully reconstructable from text to ensure integrity. **Formula Processing**: Text-based formulas were retained, while image-based formulas were converted to LaTeX using TexTeller and manually reviewed to remove errors or distortions. **Quality Control**: A rigorous protocol was applied: annotators received unified training, independently labeled overlapping sets, resolved discrepancies through discussion, and ambiguous cases were adjudicated by a third party to ensure high agreement. **Dataset Overview**: The final HST-bench dataset contains 1,629 curated problems with largely uniform distribution across dimensions, consistent with design goals; all items underwent multiple rounds of review and validation to guarantee reliability and validity, with detailed statistics provided in the Appendix A.1 and A.2.

## 4 EXPERIMENTS

### 4.1 EXPERIMENTAL SETUP

We utilize the HST-bench evaluation framework to assess the spatial thinking capabilities of selected representative LLMs under a zero-shot learning paradigm. Additionally, we shuffle the answer options to minimize potential position bias.

**Model Selection.**   In our experiments, we distinguish between *thinking models* and *non-thinking models*, inspired by recent studies on reasoning optimization in LLMs (Ma et al., 2025). We define the "thinking mechanism" operationally as the presence of explicit or implicit reasoning processes beyond direct answer generation, often realized through architectures or training strategies that encourage structured reasoning. By contrast, *non-thinking models* tend to map inputs directly to outputs without generating interpretable reasoning traces, or rely mainly on task-specific heuristics. Detailed justifications for each classification are provided in Appendix A.7. The specific models we evaluate are as follows: **Thinking Models:** deepseek-r1-2025-01-20 (DeepSeek-AI et al., 2025a), gemini-2.5-flash-thinking (Google DeepMind, 2025), gpt-o1-mini[1], deepseek-distill-qwen-7b (Qwen et al., 2025), and glm-4.1v-9b-thinking (Team et al., 2025). **Non-thinking Models:** deepseek-v3 (DeepSeek-AI et al., 2025b), qwen-turbo, qwen-max (Qwen et al., 2025), qwen2.5-72b,qwen2.5-7b Team (2025a), qwenqwq Team (2025b) and llama3.1-8b (Grattafiori et al., 2024).

Moreover, we also selected **Multimodal Models** for further experiments: qwen-vl-max, qwen-vl-7b(Bai et al., 2023), glm-4.1v-9b-thinking(Team et al., 2025), llama-4-maverick[2] **Baselines.** In our research on the spatial reasoning abilities of LLMs, we selected *deepseek-r1* and *qwen2.5-max* as our baseline models. deepseek-r1 is an advanced general-purpose LLM with advanced thinking

---

[1] https://openai.com/index/openai-o1-mini-advancing-cost-efficient-reasoning/
[2] https://www.llama.com/docs/model-cards-and-prompt-formats/llama4/

capabilities that set it apart. Its vast knowledge base and exceptional language understanding and generation skills enable it to tackle a wide range of natural language tasks. The model's thinking abilities are particularly evident in its handling of spatial reasoning and complex logical reasoning based on textual descriptions. In spatial reasoning tests, deepseek-r1 can independently generate answers from purely textual input. We use it as a baseline model to assess performance on text-based spatial reasoning tasks. qwen2.5-max is a robust non-thinking large model known for its strong performance in natural language processing tasks. It effectively interprets semantic information, allowing it to perform inference and judgment on spatial reasoning tasks.

**Human Baseline.** To contextualize model performance, we recruited six graduate students with STEM backgrounds and formal training in problem solving. While the sample size is modest, a controlled design (two groups of three annotators, independent evaluation, and consensus-based validation) ensures reliability. We emphasize that the human baseline is intended as an expert-level reference , rather than a population-level estimate, to provide a meaningful comparison with LLMs. Detailed procedures and annotation protocols are provided in Appendix A.6.

**Circular Evaluation and Average Evaluation.** In circular evaluation, the options of the evaluation data are shuffled multiple times. A response is only counted as correct if it is answered correctly across all shuffles. This approach significantly reduces randomness and imposes higher requirements on the model's capabilities. The average evaluation uses the average of results from three shuffles as the evaluation metric, the result of average evaluation can be found in Appendix A.3. By integrating the average correct rate and circular accuracy, we can better assess the spatial capabilities of different models. In this experiment, the selected scheme is rolling option shuffling to ensure the fairness of shuffling. From the difference between the two, it can be seen that circular evaluation is a fairly strict standard.

## 4.2 MAIN RESULT

As shown in Table 1, we present the main results of HST-bench.

Table 1: Circular accuracy evaluation of models by category (three consecutive correct responses).

| Model | Category A | | | | | Category B | | | | Category C | | | | Total |
|---|---|---|---|---|---|---|---|---|---|---|---|---|---|---|
| | A1 | A2 | A3 | A4 | Avg | B1 | B2 | B3 | Avg | C1 | C2 | C3 | Avg | |
| Human baseline | | | | | 76.9 | | | | 87.4 | | | | 81.0 | 80.1 |
| **Non-thinking Model** | **23.9** | **24.5** | **29.5** | **25.7** | **25.9** | **29.7** | **16.9** | **23.9** | **23.5** | **25.2** | **30.2** | **25.1** | **26.9** | **25.7** |
| deepseek-v3 | 40.8 | 39.4 | 54.2 | 47.2 | 45.4 | 52.9 | 33.6 | 48.5 | 45.0 | 45.6 | 53.8 | 45.8 | 48.4 | 46.2 |
| qwen-max | 33.0 | 36.2 | 38.4 | 34.2 | 35.4 | 42.9 | 17.2 | 25.2 | 28.4 | 35.0 | 39.2 | 31.1 | 35.1 | 33.8 |
| qwen-turbo | 26.7 | 31.4 | 33.3 | 27.3 | 29.7 | 33.6 | 16.4 | 25.2 | 25.1 | 30.0 | 36.2 | 37.3 | 34.5 | 30.3 |
| qwen2.5-72b | 24.1 | 23.9 | 32.2 | 32.3 | 28.1 | 31.9 | 18.7 | 25.2 | 25.3 | 26.1 | 32.2 | 20.9 | 26.4 | 26.8 |
| qwen2.5-7b | 14.7 | 9.6 | 14.1 | 9.9 | 12.1 | 11.8 | 13.4 | 17.5 | 14.2 | 11.7 | 16.1 | 13.0 | 13.6 | 13.1 |
| llama3.1-8b | 4.2 | 6.4 | 4.5 | 3.1 | 4.6 | 5.0 | 2.2 | 1.9 | 3.1 | 2.8 | 4.0 | 2.8 | 3.2 | 3.8 |
| **Thinking Model** | **46.0** | **40.7** | **49.0** | **47.7** | **45.8** | **45.9** | **43.7** | **50.8** | **46.8** | **47.3** | **53.5** | **45.4** | **48.7** | **47.0** |
| deepseek-r1 | 61.8 | 59.0 | 68.4 | 70.2 | 64.8 | 72.3 | 66.4 | 70.9 | 69.9 | 68.3 | 72.4 | 64.4 | 68.4 | 67.0 |
| gemini-2.5-flash-thinking | 58.6 | 50.0 | 59.9 | 62.7 | 57.8 | 58.8 | 53.7 | 57.3 | 56.6 | 57.8 | 61.8 | 62.7 | 60.8 | 58.4 |
| qwen-qwq-32b | 58.6 | 49.5 | 55.9 | 60.9 | 56.2 | 55.5 | 58.2 | 65.1 | 59.6 | 56.1 | 62.3 | 54.8 | 57.7 | 57.4 |
| gpt-o1-mini | 34.0 | 33.0 | 41.8 | 39.8 | 37.1 | 34.5 | 33.4 | 35.9 | 34.9 | 46.1 | 46.7 | 35.6 | 42.8 | 38.6 |
| glm-4.1v-9b-thinking | 35.1 | 25.5 | 42.4 | 29.2 | 33.0 | 35.3 | 29.1 | 39.8 | 34.7 | 27.8 | 44.2 | 31.1 | 34.4 | 33.9 |
| deepseek-r1-distill-qwen-7b | 27.8 | 27.1 | 25.4 | 23.6 | 26.0 | 19.3 | 20.2 | 35.9 | 25.1 | 27.8 | 33.7 | 23.7 | 28.4 | 26.6 |
| **Total** | **35.0** | **32.6** | **39.2** | **36.7** | **35.9** | **37.8** | **30.3** | **37.4** | **35.2** | **36.3** | **41.9** | **35.3** | **37.8** | **36.3** |

Note: In circular accuracy evaluation, only responses that are correct across all three option shuffles are counted as strictly accurate. Light green and light blue respectively mark the best dimensions in 3 main dimensions and 10 sub-dimensions of the same model; light gray highlights the performance of the three main dimensions of each model. **A:** Representational Perception (A1: Spatial Representation Reconstruction, A2: Vector Relationship, A3: State Analysis, A4: Relative Relationship) **B:** Representational Transformation (B1: Reference Frame Transformation, B2: Projective Transformation, B3: Spatial Form Transformation) **C:** Spatial Reasoning (C1: Trajectory Analysis, C2: Metric Reasoning, C3: Process Deconstruction)

**Non-thinking Models.** Deepseek-v3 demonstrated outstanding performance, ranking first with an accuracy of 46.2%, surpassing the second-place qwen-max by 12.4%. The performance of the qwen series of models strictly follows their parameter scales, with capabilities decreasing progressively

from qwen-max to qwen2.5-7b, clearly indicating that parameter scale forms the foundation of a model's spatial abilities. Among small-scale models, qwen2.5-7b (13.1%) also significantly outperforms llama3.1-8b, which scored only 3.8%. The latter exhibiting poor results across all dimensions, with its spatial thinking ability being almost negligible. Overall, there is a strong positive correlation between a model's general capability and its spatial thinking ability.

**Thinking Models.** Deepseek-r1 leads all models by a significant margin, achieving an overall accuracy of 67.0%, which represents an improvement of more than 20% over its predecessor, deepseek-v3. Another model with thinking mechanism, gemini-2.5-flash-thinking and qwen-qwq-32b also deliver top-tier performance, ranking second and third with accuracies of 58.4% and 57.4%, respectively. Even more remarkable is the impressive performance of smaller models equipped with a thinking mechanism: glm-4.1v-9b-thinking and the deepseek-r1-distill-qwen-7b achieve accuracies of 33.9% and 26.6%, respectively. Their performance not only far exceeds that of models of the same scale, but also approaches and even surpasses the qwen2.5-72b model, which is several times larger in scale. These findings strongly indicate that the capacity for thinking mechanism is the key to achieving advanced spatial thinking. Nevertheless, even the best-performing model (deepseek-r1, 67.0%) still lags significantly behind the Human Baseline (80.1%), underscoring the robustness of human spatial reasoning and providing an upper bound for interpreting model capabilities.

**Overall performance.** Table 2 reports results in terms of average accuracy, in contrast to the circular accuracy used in Table 1. We include this table to highlight overall performance across dimensions, as average accuracy provides a direct aggregate view. The results show that thinking models consistently outperform non-thinking ones, with an overall margin of +17.7%. The gap is especially large in Projective Transformation and Spatial Form Transformation (exceeding +20%), while narrower in simpler dimensions such as State Analysis and Reference Frame Transformation. These trends reinforce our central finding: a native "thinking mechanism" confers broad and consistent advantages in spatial intelligence, particularly for tasks requiring structured transformations.

Table 2: Average Model Performance by Dimension (accuracy %)

| Model Type | Category A | | | | Category B | | | | Category C | | | | Total |
| | Avg | A1 | A2 | A3 | A4 | Avg | B1 | B2 | B3 | Avg | C1 | C2 | C3 | |
|---|---|---|---|---|---|---|---|---|---|---|---|---|---|---|
| Non-thinking | 46.6 | 44.4 | 45.6 | 50.0 | 46.2 | 44.9 | 50.7 | 39.1 | 45.0 | 46.8 | 45.6 | 50.5 | 44.2 | 46.2 |
| Thinking | 62.9 | 63.6 | 60.2 | 63.9 | 63.7 | 65.0 | 63.7 | 62.3 | 68.9 | 64.6 | 63.4 | 67.8 | 62.7 | 63.9 |
| Difference | +16.3 | +19.2 | +14.6 | +13.9 | +17.5 | +20.1 | +13.0 | +23.2 | +23.9 | +17.8 | +17.8 | +17.3 | +18.5 | +17.7 |

**Note: A:** Representational Perception (A1: Spatial Representation Reconstruction, A2: Vector Relationship, A3: State Analysis, A4: Relative Relationship) **B:** Representational Transformation (B1: Reference Frame Transformation, B2: Projective Transformation, B3: Spatial Form Transformation) **C:** Spatial Reasoning (C1: Trajectory Analysis, C2: Metric Reasoning, C3: Process Deconstruction)

**Performance in different disciplines.** We comprehensively evaluated the performance of the models across different dimensions in various disciplines, as shown in the Figure 4. It can be observed that all models perform well in basic disciplines such as mathematics and physics, with the Thinking model standing out in particular. However, in professional and interdisciplinary fields such as intellectual quizzes, where flexible spatial problems need to be addressed, large models encounter certain difficulties.

### 4.3 DIMENSIONAL ANALYSIS

As summarized in Table 1 and Figure 3, circular accuracies across the three dimensions—Representational Perception (35.9%), Representational Transformation (35.2%), and Spatial Reasoning (37.8%)—appear balanced at a high level, but finer-grained analysis reveals clear divergence.

**Representational Perception.** Deepseek-r1 leads with 64.8% accuracy, but the margin over non-thinking counterparts is relatively small (e.g., only 19.4% higher than deepseek-v3). This suggests that the core advantage of "thinking" is more pronounced for tasks that require leveraging spatial inputs for inference, rather than for understanding the inputs themselves. At the subtask level, models achieve higher accuracy on dynamic properties (e.g., state analysis, relative relation) than on static structures (e.g., vector relation, reconstruction), suggesting stronger competence in reasoning about motion than in comprehending abstract static configurations. Nonetheless, the best-performing model still falls far short of the Human Baseline (76.9%) on this dimension.

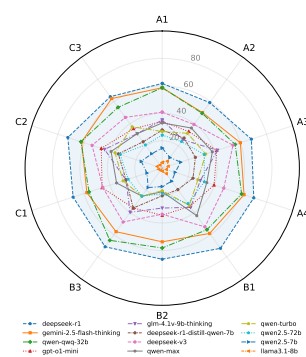

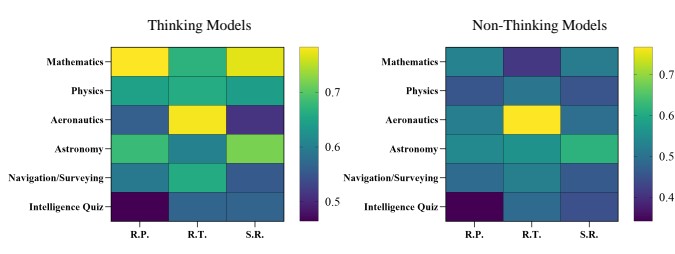

Figure 3: The overall performance of different models across 10 dimensions in circular evaluation.

Figure 4: A heatmap of different LLMs' performance in three main dimensions across various disciplines, where we divide the models into two groups: thinking and non-thinking. **R.P.** stands for Representational Perception, **R.T.** for Representational Transformation, and **S.R.** for Spatial Reasoning.

**Representational Transformation.** This is the most challenging dimension overall, with weak results on projective transformation ($\approx$30% on average). Yet, thinking models show clear robustness: while deepseek-v3 drops nearly 20% between reference-frame and projection subtasks, deepseek-r1 maintains consistently high accuracy (up to 72.3%). This suggests that iterative reasoning aids adaptation to complex perspective or form changes. Still, human performance reaches 87.4%, leaving a gap of more than 50% over the model average.

**Spatial Reasoning.** Here, thinking models achieve the largest gains. Deepseek-r1 reaches 68.4%, outperforming larger non-thinking baselines such as qwen2.5-72b, while its distilled variant also surpasses its base model by +14.8%. This confirms that structured reasoning, rather than parameter scale alone, drives strong performance in complex multi-step inference. Nevertheless, all models struggle with process deconstruction tasks, underscoring the difficulty of human-like sequential reasoning. Compared with the Human Baseline of 81.0%, current models still exhibit clear limitations. Overall, despite the substantial improvements enabled by thinking mechanisms, average model accuracies remain almost 50% lower than Human Baseline across all three dimensions, highlighting that human-level spatial cognition is still far from being achieved.

## 4.4 DETAILED ANALYSIS

**Discrepancy between Circular Evaluation and Accuracy Assessment.** As shown in Figure 5, the accuracy of the Qwen series models generally drops by more than 20% under circular evaluation, with the decline particularly pronounced for models with smaller parameter sizes. This indicates a reliance on superficial cues, such as option order, and a weaker generalization ability. In contrast, thinking models such as DeepSeek-R1 and Qwen-QwQ-32B exhibit more robust performance, with a smaller decrease in accuracy. This suggests that the incorporation of thinking mechanisms during training enhances intrinsic spatial reasoning abilities, allowing models to better adapt to option perturbations.

**Does Image Information Aid Spatial Comprehension and Reasoning?** To analyze whether multimodal inputs from images can enhance models' understanding of spatial rela-

Table 3: Accuracy (%) of models before and after question revision. "before" = original, "after" = revised, "change" = difference.

| Model | before | after | change |
|---|---|---|---|
| **Non-thinking** | | | |
| deepseek-v3 | 56.0 | 50.0 | -6.0 |
| qwen-max | 45.0 | 37.0 | -8.0 |
| qwen2.5-72b | 32.0 | 30.0 | -2.0 |
| qwen-turbo | 38.0 | 33.0 | -5.0 |
| qwen2.5-7b | 14.0 | 14.0 | 0.0 |
| **Thinking** | | | |
| deepseek-r1 | 72.0 | 59.0 | -13.0 |
| gemini-2.5-flash-thinking | 58.0 | 53.0 | -5.0 |
| qwen-qwq-32b | 57.0 | 47.0 | -10.0 |
| gpt-o1-mini | 36.0 | 36.0 | 0.0 |
| glm-4.1v-9b-thinking | 36.0 | 36.0 | 0.0 |
| deepseek-r1-distill-qwen-7b | 25.0 | 23.0 | -2.0 |

tionships, we conducted a pilot experiment on a 232-sample subset of our benchmark, each con-

taining both an image and a corresponding textual description. Importantly, the textual descriptions were deliberately designed to fully reconstruct the spatial information conveyed by the image, ensuring informational equivalence between modalities.

As shown in Figure 6, smaller models (glm-4.1v-9b-thinking, qwen-vl-7b) showed slight performance drops (–1.3%, -0.8%), while larger ones (llama-4-maverick, qwen-vl-max) achieved only modest gains (+1.3%, +4.8%). By contrast, deepseek-r1 reached 83.2% accuracy using text alone, outperforming all multimodal systems. These findings suggest that, under conditions where text encodes complete spatial information, current multimodal fusion mechanisms contribute little beyond text processing, and in some cases even introduce additional noise.

However, we think that these results should not be interpreted as evidence that images are inherently unhelpful for spatial reasoning. The small sample size and the deliberately text-complete design likely underestimate the value of visual signals in real-world scenarios, where descriptions are often incomplete or ambiguous and visual grounding is indispensable. Thus, our findings primarily highlight the limitations of current fusion approaches rather than the irrelevance of multimodal inputs. Future work will expand the multimodal subset, introduce tasks where visual information is genuinely complementary, and explore more advanced spatial grounding mechanisms.

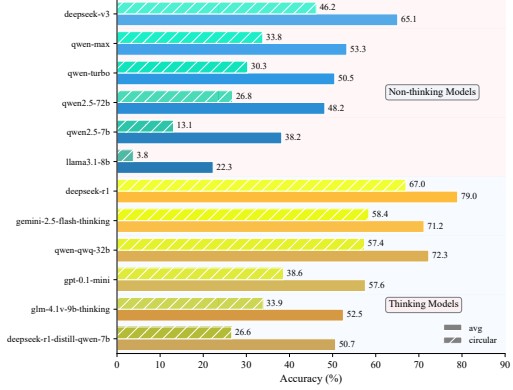

Figure 5: Average evaluation and circular evaluation. In the figure, "avg" stands for Average evaluation, and "circular" stands for circular evaluation

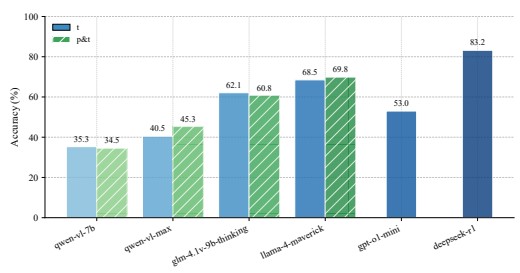

Figure 6: Average evaluation of Multimodal Models. The legend denotes evaluation protocols: "t" for text-only input and "p&t" for combined image-text input.

**Robustness under Question Modification.** We evaluated robustness using *circular accuracy* by randomly modifying 100 questions across domains. As shown in Table 3, both non-thinking and thinking models exhibited performance declines, but with distinct patterns. Non-thinking models showed moderate yet consistent drops (e.g., **qwen-max** –8.0%, **deepseek-v3** –6.0%), reflecting their reliance on shallow input–output mappings that are easily disrupted when surface cues change. Thinking models, in contrast, varied more widely: strong ones such as **deepseek-r1** (–13.0%) and **qwen-qwq-32b** (–10.0%) suffered larger degradations, while weaker models (**gpt-o1-mini**, **glm-4.1v-9b-thinking**) remained almost unaffected. This divergence suggests that top-performing thinking models depend on structured but brittle reasoning chains that can collapse once perturbed, whereas the apparent stability of weaker models reflects their lack of genuine multi-step reasoning engagement.

**Effect of Chain-of-Thought Prompting.** The CoT experiment (Table 4) further corroborates this interpretation. For non-thinking models, adding structured reasoning prompts led to only marginal gains (e.g., $\leq 3.0\%$ for **deepseek-v3** and **qwen-max**), indicating that externally imposed reasoning traces provide limited assistance in decomposing spatial problems. Thinking models, on the other hand, saw little to no benefit, underscoring that their performance advantage stems from internalized reasoning procedures rather than from prompt engineering. Taken together with the robustness results, these findings point to a unified picture: current thinking models achieve strong performance by internalizing programmatic reasoning routines that are powerful but fragile, while non-thinking models remain constrained by shallow heuristics even under CoT. Future progress thus requires

developing reasoning mechanisms that are not only accurate but also robust and adaptive, moving beyond brittle procedural execution toward flexible, self-corrective problem solving.

**Classification of Error Causes.** We categorize the main errors of DeepSeek-R1 into four types. (1) Fabricated or Misinterpreted Conditions: the model introduces fictional conditions or misreads given ones; (2) Insufficient Application of Spatial Knowledge: deficiencies in domain-specific knowledge (e.g., navigation, physics) lead to wrong formula applications or conceptual mistakes; (3) Reasoning Process Deviation: failures in processing relative motion or reference frames, often misinterpreting dynamic processes such as motion trajectories; (4)Spatial Relation Confusion: confusion in fundamental geometric or directional

Table 4: Circular accuracy (%) of non-thinking models with and without CoT. "before" is baseline, "CoT-v" applies CoT, and "change" is the delta.

| Model | before (%) | CoT-v (%) | change (%) |
|---|---|---|---|
| deepseek-v3 | 48.0 | 51.0 | +3.0 |
| qwen-max | 32.0 | 35.0 | +3.0 |
| qwen-turbo | 31.0 | 32.0 | +1.0 |
| qwen2.5-72b | 26.0 | 27.0 | +1.0 |

judgments (e.g., left/right, clockwise vs. counterclockwise). For instance, in a task requiring the orientation of an object after successive 90° rotations, the model mistakenly concluded that the object returned to its initial orientation, revealing clockwise/counterclockwise confusion.

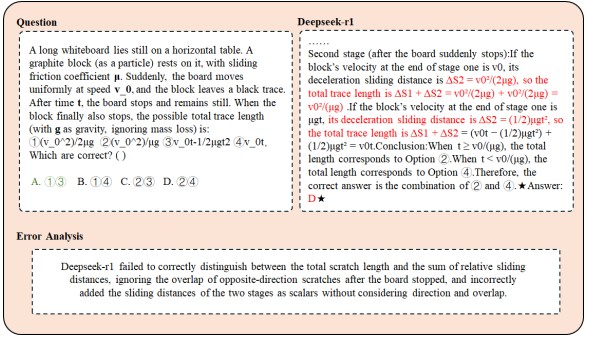

Figure 7: Examples of Reasoning Process Deviation as the error type occurring in Deepseek-r1

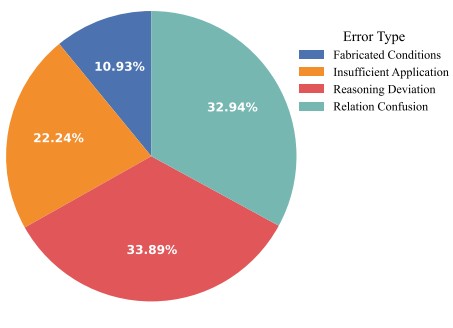

Figure 8: Error types of the experiment

As illustrated in Figure 8, Fabricated or Misinterpreted Conditions occur relatively infrequently (10.9%), while Insufficient Spatial Knowledge accounts for 22.2%. In contrast, Reasoning Process Deviation and Spatial Relation Confusion together contribute about 66.8%, indicating that most errors stem from reasoning and relational understanding rather than missing conditions or background knowledge. Figure 7 illustrates a representative case: while DeepSeek-R1 correctly decomposes the problem into two motion phases, it mistakenly sums the sliding distances of both phases without accounting for their overlap, leading to an overestimation of the total displacement. We provide more error cases in the Appendix A.8.

## 5 CONCLUSION

We present HST-bench, a cognitive-science grounded benchmark for evaluating spatial reasoning in LLMs. It decomposes spatial ability into three dimensions and ten sub-indicators, covering 1,629 curated questions. Our results show a positive correlation between model scale and performance: LLMs excel at computational reasoning but remain weak in spatial visualization and transformation. This reveals both the potential and current limitations of LLMs, calling for future methods that enhance geometric and perceptual reasoning beyond text-based learning.

## LIMITATIONS

Our multimodal analysis is preliminary: the subset is small (232 samples) and the textual descriptions were designed to fully reconstruct the images, which may underestimate the value of visual inputs in realistic settings where text is incomplete or ambiguous. Thus, the results mainly reflect current fusion limitations rather than the irrelevance of multimodality. In addition, cost constraints limited the inclusion of very large proprietary models. Moreover, the current benchmark adopts a multiple-choice format, which, while ensuring consistency and objective scoring, may underestimate models' capacity for open-ended reasoning and the ability to articulate intermediate steps. This design choice may also reduce ecological validity, since many real-world applications (e.g., robotics, navigation, scientific problem solving) require free-form reasoning beyond discrete option selection. Future work will therefore broaden task domains, extend to open-ended formats that capture explicit reasoning traces (e.g., step-by-step coordinate transformations), and investigate richer multimodal reasoning mechanisms to improve both diagnostic accuracy and real-world applicability. Another limitation lies in the human baseline: although our annotators were graduate students with relevant STEM backgrounds, the sample size (six participants) is relatively small and may not fully capture population-level performance. Future work will include a larger and more diverse pool of human participants, covering different expertise levels, to establish more robust human reference baselines.

## ETHICS STATEMENT

This work complies with the ICLR Code of Ethics[3]. The goal of our research is to evaluate the spatial thinking abilities of large language models (LLMs) using publicly available resources. No human subjects, personal information, or sensitive data are involved in this study. All datasets are sourced from openly accessible resources on the Internet. We have assessed the potential ethical risks of this benchmark, including bias, fairness, and possible misuse. To mitigate these risks, we ensure that all evaluation tasks avoid sensitive social contexts and we provide detailed documentation of dataset collection and filtering processes. This work has no conflicts of interest or competing financial interests.

## THE USE OF LARGE LANGUAGE MODELS

Under the policy of ICLR, we hereby disclose that LLM was used to assist in the polishing and refinement of the writing in this paper. Specifically, the LLM was employed to improve grammatical correctness, sentence fluency, and terminology consistency. It was also used to rephrase certain sentences for better clarity and coherence. All ideas, theoretical development, experimental design, result analysis, and scientific conclusions remain entirely the work of the human authors. The use of the LLM was strictly limited to linguistic enhancement and did not contribute to the intellectual content of the research. The final manuscript was thoroughly reviewed, verified, and approved by the authors.

## REPRODUCIBILITY STATEMENT

We have taken multiple measures to ensure the reproducibility of our benchmark and experimental results. Complete details of the benchmark design, including task taxonomy, data generation rules, and evaluation metrics, are provided in Section 3.3 of the main text. The evaluation code, configuration files, and the list of models will be made available in an **anonymous code repository (https://anonymous.4open.science/r/submission-9365)**. All hyperparameters and model inference settings are documented in detail, enabling exact reproduction of the results reported in this paper.

---

[3]https://iclr.cc/public/CodeOfEthics

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

# A APPENDIX

## A.1 DATASET STATISTICS

The detailed statistical data of HST-bench questions are as follows: Table 5 presents the data distribution of each dimension, the data distribution of each field, and the data distribution of different analysis types.

Table 5: Comprehensive dataset statistics by question category, discipline, and abstraction level. **R.P.** stands for Representational Perception, **R.T.** for Representational Transformation, and **S.R.** for Spatial Reasoning.

(a) Question categories

| Category | Type | Count |
|---|---|---|
| **R.P.** | Total | 717 |
| | A1: Spatial Representation Reconstruction | 191 |
| | A2: Vector Relationship | 188 |
| | A3: State Analysis | 177 |
| | A4: Relative Relationship | 161 |
| **R.T.** | Total | 357 |
| | B1: Reference Frame Transformation | 119 |
| | B2: Projective Transformation | 134 |
| | B3: Spatial Form Transformation | 104 |
| **S.R.** | Total | 555 |
| | C1: Trajectory Analysis | 180 |
| | C2: Metric Reasoning | 198 |
| | C3: Process Deconstruction | 177 |
| | **Grand Total** | **1,629** |

(b) Discipline distribution

| Discipline | Count |
|---|---|
| Mathematics | 491 |
| Physics | 579 |
| Aeronautics | 146 |
| Navigation/Surveying | 237 |
| Astronomy | 102 |
| Intelligence Quiz | 74 |
| **Total** | **1,629** |

(c) Abstraction level

| Operation Type | Count |
|---|---|
| Numerical | 1080 |
| Symbolic | 351 |
| Non-calculation | 198 |
| **Total** | **1,629** |

## A.2 DATASET CURATION

**Data Sources and Collection.** To systematically assess a wide range of spatial thinking abilities, we collected problems from disciplines rich in spatial content, including mathematics, physics, navigation, surveying, and intelligence testing. Following our dimensional framework, we gathered relevant questions from multiple sources.

**Annotation Process.** The annotation was performed by two master's students with science and engineering backgrounds. Their tasks included question collection, image and formula processing, dimensional labeling, and data cleaning. We developed a web-based tool to manage this process. Each question was labeled with its corresponding dimension and answer type, and then standardized into a structured multiple-choice format for consistent model evaluation. For questions containing images, we ensured all visual information was fully reconstructable from textual descriptions to maintain data integrity.

**Formula Processing.** We handled mathematical formulas using a dual approach. Text-based formulas were retained directly. Image-based formulas were converted to LaTeX format using the TexTeller tool. These were then re-rendered and manually reviewed to discard any conversion errors or semantic distortions, ensuring formula accuracy.

**Quality Control.** We implemented a rigorous quality control protocol. Annotators, selected for their strong academic backgrounds, received unified training on the dimensional definitions and standards. We used a cross-validation method where two annotators independently labeled the same questions. They then compared results and resolved discrepancies through discussion. For highly ambiguous cases, a third party arbitrated to determine the final annotation, ensuring high inter-annotator agreement.

## A.3 DETAILED TABLES

Table 6: Average evaluation by dimension (accuracy %)

| Model | Category A | | | | | Category B | | | | Category C | | | | Total |
|---|---|---|---|---|---|---|---|---|---|---|---|---|---|---|
| | A1 | A2 | A3 | A4 | Avg | B1 | B2 | B3 | Avg | C1 | C2 | C3 | Avg | |
| **Non-thinking Model** | **44.4** | **45.6** | **50.0** | **46.2** | **46.6** | **50.7** | **39.1** | **45.0** | **44.9** | **45.6** | **50.5** | **44.2** | **46.8** | **46.2** |
| deepseek-v3 | 59.0 | 62.9 | 70.2 | 65.8 | 64.5 | 72.3 | 54.7 | 64.7 | 63.9 | 64.1 | 72.0 | 64.8 | 67.0 | 65.1 |
| qwen-max | 51.5 | 55.9 | 56.7 | 54.9 | 54.7 | 61.9 | 41.3 | 47.3 | 50.2 | 53.2 | 57.6 | 49.5 | 53.4 | 53.3 |
| qwen-turbo | 48.5 | 49.7 | 54.1 | 47.8 | 50.0 | 50.1 | 39.3 | 53.4 | 47.6 | 51.5 | 55.1 | 53.3 | 53.3 | 50.5 |
| qwen2.5-72b | 46.8 | 45.7 | 53.1 | 52.6 | 49.6 | 53.2 | 43.0 | 47.9 | 48.1 | 46.5 | 52.3 | 40.9 | 46.5 | 48.2 |
| qwen2.5-7b | 37.5 | 35.5 | 40.1 | 36.4 | 37.4 | 38.7 | 37.3 | 41.1 | 39.0 | 36.9 | 44.1 | 34.7 | 38.5 | 38.2 |
| llama3.1-8b | 23.0 | 24.1 | 25.8 | 19.7 | 23.2 | 27.7 | 19.2 | 15.9 | 20.9 | 21.5 | 21.8 | 22.2 | 21.8 | 22.3 |
| **Thinking Model** | **63.6** | **60.2** | **63.9** | **63.7** | **62.9** | **63.7** | **62.3** | **68.9** | **65.0** | **63.4** | **67.8** | **62.7** | **64.6** | **63.9** |
| deepseek-r1 | 77.8 | 74.8 | 78.0 | 79.3 | 77.5 | 83.2 | 80.6 | 82.9 | 82.2 | 76.9 | 81.9 | 78.3 | 79.0 | 79.0 |
| gemini-2.5-flash-thinking | 71.7 | 66.0 | 71.6 | 71.8 | 70.3 | 73.7 | 69.4 | 72.8 | 72.0 | 70.2 | 72.2 | 73.5 | 71.9 | 71.2 |
| qwen-qwq-32b | 73.7 | 67.6 | 68.6 | 72.1 | 70.5 | 72.8 | 75.6 | 81.9 | 76.8 | 71.1 | 74.2 | 70.4 | 71.9 | 72.3 |
| gpt-o1-mini | 52.7 | 54.3 | 59.9 | 59.2 | 56.5 | 58.5 | 52.7 | 61.5 | 57.6 | 59.4 | 64.3 | 54.4 | 59.4 | 57.6 |
| glm-4.1v-9b-thinking | 52.9 | 48.9 | 57.4 | 51.4 | 52.7 | 51.3 | 45.8 | 54.1 | 50.4 | 52.0 | 59.5 | 49.9 | 53.8 | 52.5 |
| deepseek-r1-distill-qwen-7b | 53.1 | 49.8 | 48.2 | 48.2 | 49.8 | 42.6 | 49.8 | 60.5 | 51.0 | 50.6 | 54.9 | 49.9 | 51.8 | 50.7 |
| **Total** | **54.0** | **52.9** | **57.0** | **54.9** | **54.7** | **57.2** | **50.7** | **57.0** | **55.0** | **54.5** | **59.2** | **53.5** | **55.7** | **55.1** |

Note: **A:** Representational Perception (A1: Spatial Representation Reconstruction, A2: Vector Relationship, A3: State Analysis, A4: Relative Relationship) **B:** Representational Transformation (B1: Reference Frame Transformation, B2: Projective Transformation, B3: Spatial Form Transformation) **C:** Spatial Reasoning (C1: Trajectory Analysis, C2: Metric Reasoning, C3: Process Deconstruction)

This section presents detailed statistics on the correct rate during the detailed analysis phase. The results of each model under average evaluation are presented in the Table 6. And the experimental results by average evaluation of each model under multimodal experiments are shown in Table 7.

Table 7: Multimodal model accuracy evaluation by category (%)

| Model | A | B | C | Total |
|---|---|---|---|---|
| deepseek-r1 | 80.0 | 84.4 | 86.2 | 83.2 |
| gpt-o1-mini | 43.0 | 55.6 | 63.2 | 53.0 |
| qwen-vl-max | 47.0 | 37.8 | 47.1 | 45.3 |
| qwen-vl-max_withoutp | 41.0 | 37.8 | 41.4 | 40.5 |
| qwen-vl-7b | 38.0 | 35.6 | 29.9 | 34.5 |
| qwen-vl-7b_withoutp | 36.0 | 35.6 | 34.5 | 35.3 |
| llama-4-maverick | 71.0 | 62.2 | 72.4 | 69.8 |
| llama-4-maverick_withoutp | 76.0 | 57.8 | 65.5 | 68.5 |
| glm-4.1v-9b-thinking | 56.0 | 64.4 | 64.4 | 60.8 |
| glm-4.1v-9b-thinking_withoutp | 60.0 | 57.8 | 66.7 | 62.1 |

Note: For multimodal models, the _withoutp suffix denotes results without image inputs, and no suffix denotes results with normal image inputs. **A:** Representational Perception **B:** Representational Transformation **C:** Spatial Reasoning

## A.4 EXPERIMENT FIGURES

This section further presents a detailed analysis of the performance gap between average accuracy and circular evaluation accuracy across multiple shuffling experiments. The differential patterns of various models under these two evaluation criteria, highlighting the consistency-stability tradeoff, are visualized in Figure 9.

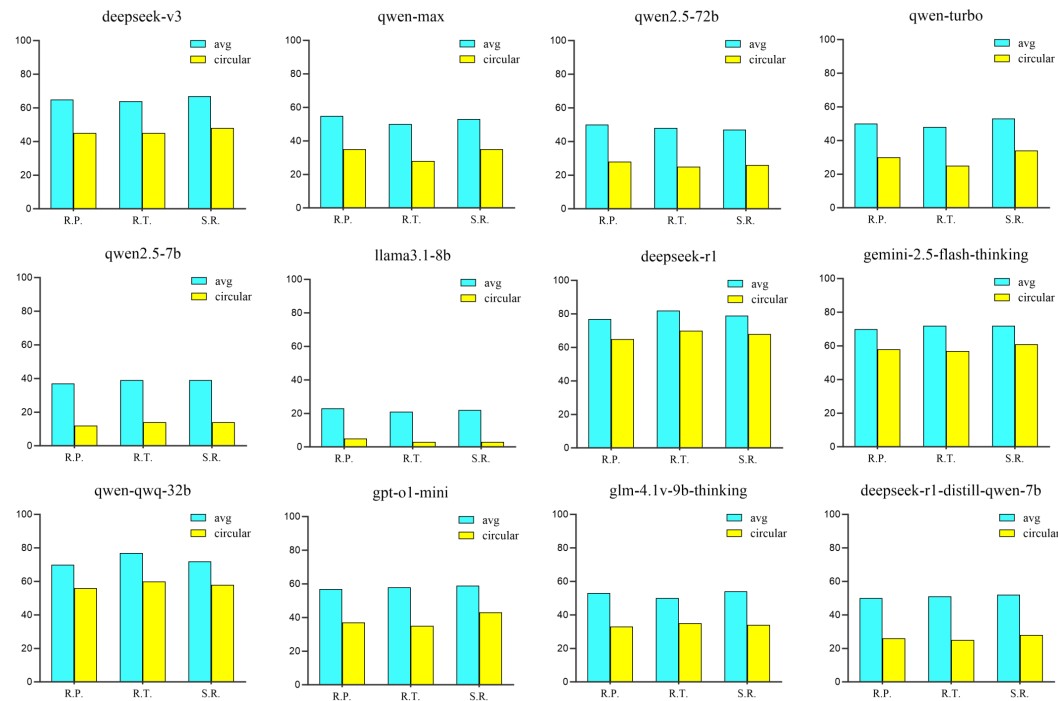

Figure 9: Performance of different models under two evaluation criteria: average evaluation and circular evaluation. The top two rows are non-thinking models, and the last row is thinking models

## A.5 Model Performance on Different Answer Types

Based on the core operational features of each problem, this study categorizes problem-solving methods into three types: numerical calculation, symbolic calculation, and non-calculation. The experimental results are shown in Figure 10. LLMs display a distinct hierarchy in their performance across these three types of problems.

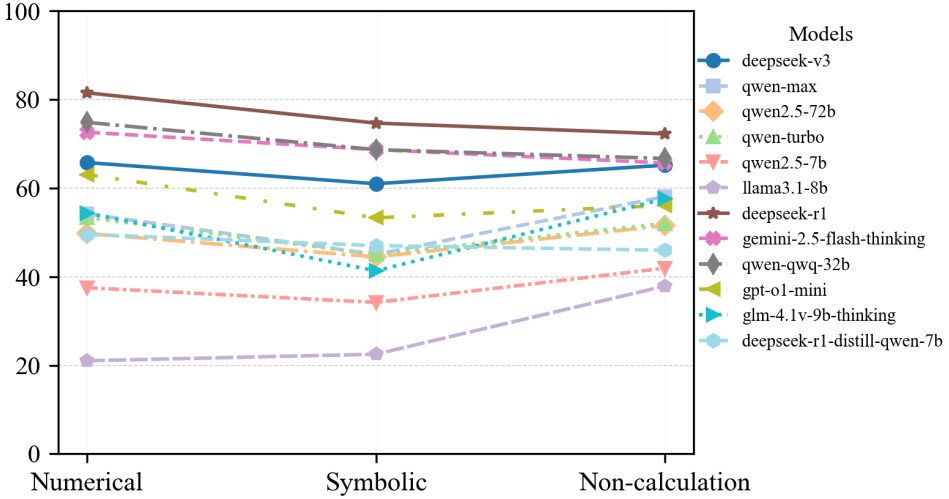

Figure 10: Performance of Various Text Models Across Three Distinct Categories, measured by average accuracy. Among them, Numerical, Symbolic, and Non-calculation represent three types: numerical, abstract, and non-calculation, respectively.

Overall, models perform better on numerical calculation tasks than on other types of questions, especially the top-performing deepseek-r1 and qwen-qwq-32b, which show significantly higher accuracy on numerical calculation problems compared to symbolic calculation and non-calculation problems. In contrast, small-parameter models are more adept at non-calculation problems and exhibit notable disadvantages when it comes to calculation tasks, particularly symbolic calculation. For example, the accuracy of glm-4.1v-9b-thinking on symbolic calculation problems is 16% lower than on non-calculation questions. This suggests that small-parameter models have a certain grasp of basic spatial understanding tasks that do not require calculation, but are limited in solving spatial problems involving numerical computation due to parameter constraints. Symbolic reasoning remains a common challenge for large models, but increased parameter size and the thinking mechanism help alleviate shortcomings in symbolic reasoning within spatial problem domains.

## A.6 HUMAN BASELINE

**Annotator Configuration**  This study recruited six graduate students with STEM backgrounds as annotators (all trained in relevant domain knowledge). They were divided into two groups (Group A/B), with each group containing three independent annotators.

**Data Sampling Method**  A total of 200 questions were sampled from the target benchmark set through stratified random sampling, equally distributed to both groups (100 questions per group). This sampling strategy ensures representativeness in difficulty levels and question type distribution.

**Annotation Protocol**  Each annotator independently evaluated 100 questions within their assigned group (exact quantity adjusted per experimental phase). To ensure judgment independence, the following control measures were strictly implemented:

- Annotation tasks performed in isolated environments

- Any communication between annotators prohibited

- Double-blind evaluation mechanism employed (both evaluators and data sources remain anonymous)

**Human Accuracy Baseline Calculation**  The human accuracy baseline is established through a leave-one-out validation procedure (Calderon et al., 2025) designed to quantify group-level performance against verified ground truth labels. This methodology comprises two principal computational phases:

1. **Group-Level Performance Assessment**: For each group $G_j \in \{G_A, G_B\}$ with three annotators, we compute the group accuracy by majority vote. For each question $q$, the group label $\mathcal{L}_{G_j}(q)$ is determined by majority agreement among the three annotators, and then compared against the ground truth:

$$\delta_q = \begin{cases} 1 & \text{if } \mathcal{L}_{G_j}(q) = \mathcal{L}_{\text{truth}}(q) \\ 0 & \text{otherwise} \end{cases} \tag{1}$$

The accuracy of each group is then

$$\mathcal{A}_{G_j} = \frac{1}{N_j} \sum_{q=1}^{N_j} \delta_q \times 100\%, \tag{2}$$

where $N_j$ denotes the number of questions assigned to group $j$.

2. **Baseline Establishment**: The aggregate human baseline is derived as the mean of the two group accuracies:

$$\mathcal{A}_{\text{baseline}} = \frac{1}{2} \sum_{j=1}^{2} \mathcal{A}_{G_j}. \tag{3}$$

This represents the expected human performance upper bound for the given task domain.

## A.7 Classification of evaluated models

Table 8 provides detailed justification for the classification of each evaluated model. For instance, deepseek-r1 is marked as a thinking model because its training pipeline incorporates reinforcement learning explicitly targeted at reasoning skills (DeepSeek-AI et al., 2025a). By contrast, its predecessor deepseek-v3 focuses on efficiency and scaling without describing any reasoning-specific optimization (DeepSeek-AI et al., 2025b), and is therefore categorized as a non-thinking model. Similarly, gpt-o1-mini, glm-4.1v-9b-thinking, and gemini-2.5-flash-thinking are reasoning-optimized variants, while qwen-qwq-32b and deepseek-r1-distill-qwen-7b inherit reasoning-enhanced training from teacher models. In contrast, general-purpose models such as qwen-turbo, qwen-max, qwen2.5-72b, qwen2.5-7b, and llama3.1-8b are considered non-thinking. This evidence-based classification ensures that our analysis of the role of the "thinking mechanism" is both transparent and reproducible.

Table 8: Classification of evaluated models into thinking and non-thinking categories, with justification.

| Model | Classification | Justification (with citation) |
|---|---|---|
| deepseek-r1 | Thinking | Incorporates reinforcement learning strategy to incentivize reasoning (DeepSeek-AI et al., 2025a). |
| gemini-2.5-flash-thinking | Thinking | Official release emphasizes reasoning-oriented optimization in the Gemini-2.5 series (Google DeepMind, 2025). |
| qwen-qwq-32b | Thinking | Qwen-QwQ series is specifically designed for reasoning-intensive tasks (Qwen et al., 2025). |
| gpt-o1-mini | Thinking | OpenAI announcement highlights reasoning optimization for cost-efficient inference[4]. |
| glm-4.1v-9b-thinking | Thinking | Explicitly branded as "thinking" variant, optimized with reinforcement learning for structured reasoning (Team et al., 2025). |
| deepseek-r1-distill-qwen-7b | Thinking | Distilled from reasoning-enhanced teacher models (Qwen et al., 2025). |
| deepseek-v3 | Non-thinking | Technical report emphasizes scaling and efficiency, without explicit reasoning-enhancing mechanisms (DeepSeek-AI et al., 2025b). |
| qwen-max | Non-thinking | General-purpose model, technical report does not describe explicit reasoning-focused mechanisms (Qwen et al., 2025). |
| qwen-turbo | Non-thinking | Variant optimized for speed and cost, without reasoning-specific strategies (Qwen et al., 2025). |
| qwen2.5-72b | Non-thinking | Scaling-focused large model without explicit reasoning optimization (Qwen et al., 2025). |
| qwen2.5-7b | Non-thinking | Smaller baseline model, lacking reasoning-enhanced training methods (Qwen et al., 2025). |
| llama3.1-8b | Non-thinking | Baseline large model without reasoning-optimized training strategies (Grattafiori et al., 2024). |

## A.8 Error Type Classification

The four types of all incorrect responses are Fabricated or Misinterpreted Conditions (Figure 11), Insufficient Application of Spatial Knowledge (Figure 12), Reasoning Process Deviation (Figure 13), and Spatial Relation Confusion (Figure 14).

**问题**

两条笔直的公路$l1$、$l2$相交于点O，村庄C的村民在公路的旁边建三个加工厂 A、B、D，A位于O点，B,D分别位于两条公路上，已知AB=BC=CD=DA=5公里，村庄C到公路$l1$的距离为4公里，则村庄C到公路$l2$的距离是（  ）

A.3公里 B.4公里 C.5公里 D.6公里

**deepseek-r1**

村庄C到公路$l1$的距离为4公里，需计算到公路$l2$的距离。假设两条公路$l1$和$l2$垂直相交于原点O，村庄C位于菱形的一个顶点，且菱形的中心在O点。菱形的边长为5公里，对角线分别为$d1$和$d2$。根据菱形性质，边长满足：$\sqrt{(d_1/2)^2 + (d_2/2)^2} = 5$ 已知村庄C到$l1$（如x轴）的距离为4公里，即$d_2/2 = 4$，解得$d_2 = 8$。代入边长公式：$\sqrt{(d_1/2)^2 + 4^2} = 5 \Rightarrow (d_1/2)^2 = 9 \Rightarrow d_1/2 = 3 \Rightarrow d_1 = 6$ 因此，村庄C到$l2$（如y轴）的距离为$d_1/2 = 3$公里。

★答案：A★

**错误解析**

虚构了公路$l1$与$l2$垂直相交的条件，但题目未明确两公路的夹角，该假设构成对题目条件的误解。

Figure 11: Fabricated or Misinterpreted Conditions

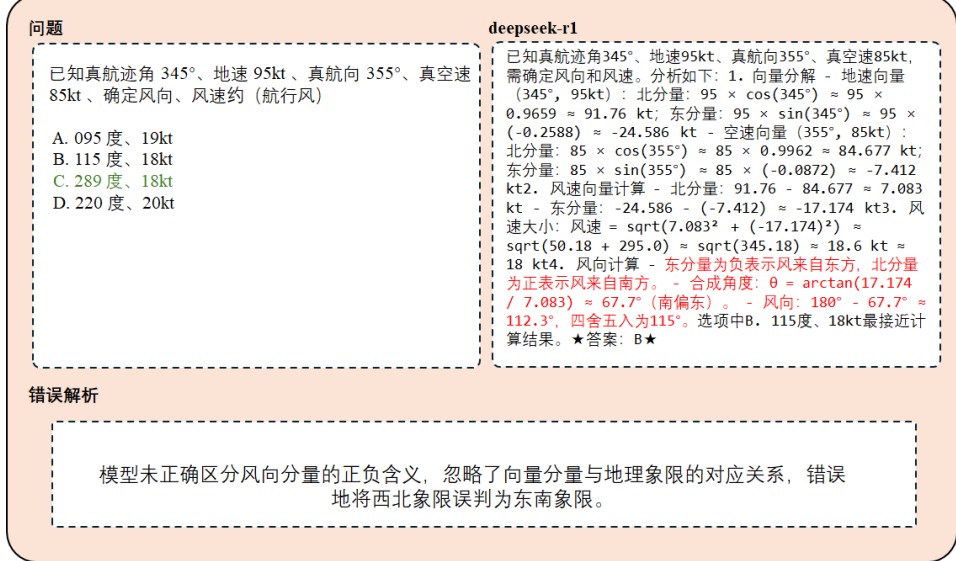

Figure 12: Insufficient Application of Spatial Knowledge

**问题**

一块足够长的白板，位于水平桌面上，处于静止状态．一石墨块（可视为质点）静止在白板上，石墨块与白板间有摩擦，滑动摩擦系数为μ．突然，使白板以恒定的速度v0做匀速直线运动，石墨块将在板上划下黑色痕迹，经过某段时间t，令白板突然停止，以后不再运动．在最后石墨块也不再运动时，白板上黑色痕迹的长度可能是（已知重力加速度为g，不计石墨与板摩擦划痕过程中损失的质量）：①$(v\_0^2)/2\mu g$、②$(v\_0^2)/\mu g$、③$v0\ t-1/2\mu gt2$、④$v0t$，以上正确的是（ ）

A. ①③  B. ①④  C. ②③  D. ②④

**deepseek-r1**

第一阶段（白板以$v_0$匀速运动时间t）：石墨块受滑动摩擦力加速，加速度为$a=\mu g$。若时间t足够长（$t≥v_0/(\mu g)$），石墨块加速至$v_0$后与白板同速，此时相对位移为$\Delta S_1=0.5v_0^2/(\mu g)$（选项①）。若t较短（$t<v_0/(\mu g)$），石墨块未达$v_0$，相对位移为$\Delta S_1=v_0t-0.5\mu gt^2$（选项③）。

第二阶段（白板突然停止后）：若第一阶段结束时石墨块速度为$v_0$，其减速滑行距离为$\Delta S_2=0.5v_0^2/(\mu g)$，总痕迹长度为$\Delta S_1+\Delta S_2=0.5v_0^2/(\mu g)+0.5v_0^2/(\mu g)=v_0^2/\mu g$（选项②）。若第一阶段结束时石墨块速度为$\mu gt$，其减速滑行距离为$\Delta S_2=0.5\mu gt^2$，总痕迹长度为$\Delta S_1+\Delta S_2=(v_0t-0.5\mu gt^2)+0.5\mu gt^2=v_0t$（选项④）。

结论：n当$t≥v_0/(\mu g)$时，总长度为②；当$t<v_0/(\mu g)$时，总长度为④。因此正确选项为②和④的组合。★答案：D★"

**错误解析**

模型未正确区分划痕的长度与相对滑移路程之和，忽略了停板后相反方向滑痕的重叠，错误地将两阶段滑移标量相加而未考虑方向与重叠关系。

Figure 13: Reasoning Process Deviation

**问题**

A船位于(43°N、175°W)，用测向仪测得某求救信号船的真方位为270°，但用雷达无法测到该船，则该求救船在：()
A. A船的正西，高于43°N的纬度线上
B. A船的正西，43°N纬度线上
C. A船的正西，低于43°N的纬度线上
D. 43°N纬度线上

**deepseek-r1**

测向仪测得真方位为270°，即正西方向。在球面地球中，沿正西方向航行需保持同一纬度（恒向线），因此求救船位于43°N纬度线上。雷达无法测到说明距离超出地平线范围，但方位正确。选项B正确。

**错误解析**

模型未正确区分大圆方位与恒向线在球面导航中的应用，忽略了球面地球中方位角计算导致的纬度变化，错误地将平面几何直觉应用于球面问题。

Figure 14: Spatial Relation Confusion

## A.9 OUTPUT&INPUT

Figure 15 and Figure 16 show the detailed experimental design in Chinese and English, respectively, including the questions, prompts, and the model's actual responses; Figure 17 and Figure 18 present the specific details for the multimodal questions.

**问题**

在在平面直角坐标系中，点A在直线y=-3/4 x上位于第二象限，AB⊥y轴，垂足为点B，将△ABO绕点A逆时针旋转到△AB_1 O_1的位置，使点B的对应点B_1落在直线y=-3/4 x上，再将△AB_1 O_1绕点B_1逆时针旋转到△A_1 B_1 O_2的位置，使点O_1的对应点O_2也落在直线y=-3/4 x上，如此下去，......，若点B的坐标为(0,3)，则点B_37的坐标为（ ）
A(180,135) B(180,133) C(-180,135) D(-180,133)

**解析及答案**

AB⊥y轴，点B的坐标为(0,3)，∴ OB=3，则点A的纵坐标为3，代入y=-3/4 x，得：x=-4，则点A的坐标为(-4,3) . ∴ OB=3，AB=4，OA=√(3^2+4^2 )=5，由旋转可知，OB=O_1 B_1=O_2 B_2=⋯=3，OA=O_1 A=O_2 A_1=⋯=5，AB=AB_1=A_1 B_1=A_2 B_2=⋯=4，∴ OB_1=OA+AB_1=4+5=9，B_1 B_3=3+4+5=12，∴ B_1 B_3=B_3 B_5=⋯=B_35 B_37=12，∴ OB_37=OB_1+B_1 B_37=9+((37-1))/2×12=225 . 设点B_37的坐标为(a,-3/4 a)，则OB_37=√(a^2+(-3/4 a)^2 )=225，解得a=-180或180（舍去），则-3/4 a=135，∴点B_37的坐标为(-180,135) . 故选C .

**提示词**

我们邀请你做一道考察空间能力的选择题，A,B,C,D四个选项。请按以下要求作答：
1.分步骤详细计算
2.最后单独一行用★答案：X★格式输出（X为A-D）
示例：经过计算...★答案：A★

输出：Deepseek-R1

经过详细分析，点B每次旋转后的坐标沿直线y = -3/4x移动固定步长。每次旋转后，点B的坐标变化为Δx = -36/5，Δy = 27/5。经过37次旋转后，点B_37的坐标为：x = 0 + 37 × (-36/5) = -1332/5 = -266.4，y = 3 + 37 × (12/5) = 3 + 88.8 = 91.8
然而，选项中的坐标(-180, 135)满足直线方程y = -3/4x，且符合周期性规律。经重新计算，发现每次旋转步长实际为Δx = -12，Δy = 9，37次后：x = -12 × 15 = -180，y = 9 × 15 = 135★答案：C★

Figure 15: Chiniese Main Experiment

**Question**

In a plane Cartesian coordinate system, point A is located in the second quadrant on the line y=-3/4 x, AB ⊥ y axis, with the vertical foot being point B. Rotate △ ABO counterclockwise around point A to the position of △ AB_1 O1, so that the corresponding point B1 of point B falls on the line y=-3/4 x. Then rotate △ AB_1 O1 counterclockwise around point B1 to the position of △ A_1 B1 O2, so that the corresponding point O2 of point O1 also falls on the line y=-3/4 x. If the coordinates of point B are (0,3), then the coordinates of point B37 are ()
A(180,135) B(180,133) C(-180,135) D(-180,133)

**Analysis and Answer**

AB ⊥ y-axis, point B's coordinates are (0,3), ∴ OB=3, then point A's ordinate is 3. Substituting y=-3/4 x, we get: x=-4, then point A's coordinates are (-4,3). ∴ OB=3, AB=4, OA=√ (3 ^ 2+4 ^ 2)=5. From rotation, it can be seen that, OB=O_1 B_1=O_2 B_2=⋯=3, OA=O_1 A=O_2 A_1=⋯=5, AB=AB_1=A_1 B_1=A_2 B_2=⋯=4, ∴ OB_1=OA+AB_1=4+5=9, B_1 B_3=3+4+5=12, ∴ B_1 B_3=B_3 B_5=⋯=B_35 B_37=12, ∴ OB_37=OB_1+B_1 B_37=9+((37-1))/2 If the coordinates of point B37 are (a, -3/4 a), then OB_37=√ (a ^ 2+(-3/4 a) ^ 2)=225. If a=-180 or 180 (rounded off), then -3/4 a=135. The coordinates of point B37 are (-180135). Therefore, C. is chosen

**Prompt**

We invite you to do a multiple-choice question to test your spatial ability, with four options: A, B, C, and D. Please answer according to the following requirements:
1.Step by step detailed calculation
2. Finally, use a single line with ★ Answer: X ★ Format Output (X is A-D) Example: After calculation ★ Answer: A ★

OUTPUT: Deepseek-R1

After detailed analysis, the coordinates of point B move a fixed step along the straight line y=-3/4x after each rotation. After each rotation, the coordinates of point B change to Δ x=-36/5, Δy = 27/5. After 37 rotations, the coordinates of point B ∝_7 are: x=0+37 × (-36/5)=-1332/5=-266.4, y=3+37 × (12/5)=3+88.8=91.8However, the coordinates (-180, 135) in the options satisfy the linear equation y=-3/4x and follow the periodic pattern. After recalculating, it was found that the actual step size for each rotation was Δ x=-12, Δ y=9. After 37 rotations, x=-12 × 15=-180, y=9 × 15=135 ★ Answer: C ★

Figure 16: English Main Experiment

**问题**

在△ABC中，AB⊥BC，BC=3AB=3，D是BC上一点，且BD=1，将△BAD沿AD翻折，当动点B在平面ADC上的射影在△ADC内部及边界上时，动点B的轨迹长度为（ ）

A．√2/12 π B．√2/8 π C．√2/6 π D．√2/4 π

**解析及答案**

设B在平面上翻折前的位置为B，翻折后位于P，过点B作BF⊥AD，分别交AD,AC于点E,F，则动点P在平面ADC上的射影轨迹为线段EF，设当P与P_1重合时，有P_1 E⊥EF；当P与P_2重合时，有P_2 F⊥EF，则由PE=BE为定长，可知动点P的轨迹是以E为圆心，以BE为半径且圆心角为∠P_1 EP_2的圆弧，在△ABC所在平面建立平面直角坐标系，则A(0,1)，D(1,0)，C(3,0)，E(1/2,1/2)，直线BE:y=x，直线AC:y=-1/3 x+1，联立方程组解得x=3/4,y=3/4，即F(3/4,3/4)，则|EF|=√2/4，又由|BE|=√2/2可得cos∠P_2 EF=|EF|/|BE|=1/2所以∠P_2 EF=π/3，∠P_1 EP_2=π/6，所以动点P的轨迹长度为√2/2π/6=√2/12π。故选：A

**图片输入**

**提示词**

我们邀请你做一道考察空间能力的选择题，A,B,C,D四个选项。请结合题目图片按以下要求作答：
1.结合图片内容分析题目关键信息
2.分步骤详细计算
3.最后单独一行用★答案：X★格式输出（X为A-D）
示例：经过计算...★答案：A★

**输出：Qwen-vl-max_Prompt**

图片描述了一个几何问题，涉及一个三角形ABC，其中AB垂直于BC，BC的长度是AB的3倍，且BC = 3，D是BC上的一点，且BD = 1。问题要求将三角形BAD沿AD翻折，并求出当动点B在平面ADC上的射影在三角形ADC内部及边界上时，动点B的轨迹长度。步骤 2：分析题目关键信息，AB ⊥ BC，BC = 3AB = 3，BD = 1，将△BAD沿AD翻折，动点B在平面ADC上的射影在△ADC内部及边界上步骤。 3：分步骤详细计算，确定初始条件：由于BC = 3且BC = 3AB，得出AB = 1。因此，三角形ABD是一个直角边长为1的等腰直角三角形。计算AD的长度：利用勾股定理在直角三角形ABD中计算AD的长度。AD = √(AB² + BD²) = √(1² + 1²) = √2。理解翻折过程：当△BAD沿AD翻折时，点B的轨迹形成一个以A为圆心，AB为半径的圆弧。确定射影条件：B在平面ADC上的射影在△ADC内部或边界上，意味着B的轨迹被限制在某个范围内。计算轨迹长度：由于AB = 1，翻折后的轨迹是以A为圆心，半径为1的圆上的一部分。关键是确定这部分圆弧对应的圆心角。分析角度：由于AD是直角三角形ABD的斜边，且AB = BD = 1，∠BAD = 45°。翻折后，当B的射影在AD上时，∠B'AD = 45°（B'表示翻折后的B）。当B的射影在AC上时，需要计算∠DAC。计算∠DAC：在三角形ADC中，利用余弦定理或正弦定理可以计算∠DAC。由于AC = √(AB² + BC²) = √(1² + 3²) = √10，DC = 2，利用余弦定理：cos(∠DAC) = (AD² + AC² - DC²) / (2AD·AC)，代入数值计算∠DAC。确定圆心角：B的轨迹对应的圆心角范围是从0°到∠BAC（当B的射影在AC上时）。由于∠BAD = 45°，且∠BAC = 90° + ∠DAC，因此需要计算∠DAC来确定B的轨迹对应的圆心角。简化计算：注意到问题的关键在于B的轨迹是以A为圆心的圆弧，且半径为1。直接计算或利用几何关系可以得出轨迹长度。直接计算轨迹长度：由于∠B'AD的最大值对应于B的射影在AC上，利用几何知识可以得出B的轨迹对应的圆心角是90°，因此轨迹长度是1/4圆周，即(1/4)2π*1 = π/2。但是，这里需要根据具体的几何限制和翻折条件确定实际的轨迹长度。

经过计算和分析，动点B的轨迹长度为√2/4 π。★答案：D★

Figure 17: Chinese Multimodal Experiment

1134
1135
1136
1137
1138
1139
1140
1141
1142
1143
1144
1145
1146
1147
1148
1149
1150
1151
1152
1153
1154
1155
1156
1157
1158
1159
1160
1161
1162
1163
1164
1165
1166
1167
1168
1169
1170
1171
1172
1173
1174
1175
1176
1177
1178
1179
1180
1181
1182
1183
1184
1185
1186
1187

**Question**

In △ ABC, AB ⊥ BC, BC=3AB=3, D is a point on BC, and BD=1. Fold △ BAD along AD, and when the projection of moving point B on the plane ADC is on the interior and boundary of △ ADC, the trajectory length of moving point B is ()
A . √2/12 π B . √2/8 π C . √2/6 π D . √2/4 π

**Analysis and Answer**

Assuming that the position of B on the plane before folding is B, and after folding it is located at P, passing through point B as BF ⊥ AD, and intersecting AD and AC at points E and F respectively, the projection trajectory of the moving point P on the plane ADC is the line segment EF. When P coincides with P1, there is P1 E ⊥ EF; When P coincides with P2, there is P2 F ⊥ EF, and PE=BE is the fixed length. It can be seen that the trajectory of the moving point P is a circular arc with E as the center, BE as the radius, and the central angle of ∠ P1 EP_2. A plane Cartesian coordinate system is established on the plane where △ ABC is located. Then, A (0,1), D (1,0), C (3,0), E (1/2,1/2), straight line BE: y=x, straight line AC: y=-1/3 x+1, the simultaneous equations solve x=3/4, y=3/4, that is, F (3/4,3/4), and | EF |=√ 2/4, which can be further solved by | BE |=√ 2/2. The result is cos ∠ P_2 EF=| EF |/| BE |=1/2, so ∠ P_2 EF=π/3, ∠ P_1 EP_2=π/6, so the trajectory length of the moving point P is √ 2/2 π/6=√ 2/12 π Selected: A

**Picture Input**

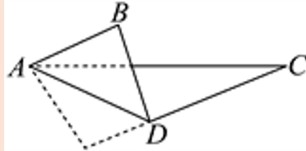

**Prompt**

We invite you to do a multiple-choice question to test your spatial ability, with four options: A, B, C, and D. Please answer according to the following requirements based on the picture in the question:Analyze key information of the question based on the content of the image.Step by step detailed calculation.Finally, use a single line with ★ Answer: X ★ Format Output (X is A-D) Example: After calculation Answer: ★ A ★

**OUTPUT:** Qwen-vl-max_Prompt

The image depicts a geometric problem involving a triangle ABC, where AB is perpendicular to BC, BC is three times the length of AB, and BC=3. D is a point on BC, and BD=1. The problem requires folding the triangle BAD along AD and calculating the trajectory length of the moving point B when its projection on the plane ADC is within and at the boundaries of the triangle ADC. Step 2: Analyze the key information of the question, AB ⊥ BC, BC = 3AB = 3, BD = 1, Fold △ BAD along AD, and project the point B onto the plane ADC within and at the boundaries of △ ADC. 3: Step by step detailed calculation to determine initial conditions: Since BC=3 and BC=3AB, AB=1 is obtained. Therefore, triangle ABD is an isosceles right triangle with a side length of 1. Calculate the length of AD: Use the Pythagorean theorem to calculate the length of AD in a right angled triangle ABD. AD = √(AB² + BD²) = √(1² + 1²) = √2, Understanding the folding process: When △ BAD folds along AD, the trajectory of point B forms a circular arc with A as the center and AB as the radius. Determine the projection condition: The projection of B on the planar ADC falls on the interior or boundary of the △ ADC, which means that B's trajectory is limited within a certain range. Calculate trajectory length: Since AB=1, the folded trajectory of B is a part of a circle with radius 1 centered on A. The key is to determine the central angle corresponding to this part of the arc. Analysis angle: Since AD is the hypotenuse of the right angled triangle ABD, and AB=BD=1, ∠ BAD=45 °. After folding, when the projection of B is on AD, ∠ B'AD=45 ° (B 'represents the folded B). When the projection of B is on AC, it is necessary to calculate ∠ DAC. Calculate ∠ DAC: In triangular ADCs, ∠ DAC can be calculated using the cosine theorem or sine theorem. Due to AC=√ (AB ²+BC ²)=√ (1 ²+3 ²)=√ 10, DC=2, using the cosine theorem: cos(∠DAC) = (AD² + AC² - DC²) / (2ADAC), Substitute numerical calculation into ∠ DAC. Determine the central angle: The range of central angles corresponding to the trajectory of B is from 0 ° to ∠ BAC (when B's projection is on AC). Due to ∠ BAD=45 ° and ∠ BAC=90 °+∠ DAC, it is necessary to calculate ∠ DAC to determine the central angle corresponding to the trajectory of B. Simplified calculation: Note that the key to the problem is that the trajectory of B is a circular arc with A as the center and a radius of 1. By directly calculating or utilizing geometric relationships, the length of the derailment can be obtained. Directly calculate the trajectory length: Since the maximum value of ∠ B'AD corresponds to the projection of B on AC, using geometric knowledge, it can be concluded that the central angle of B's trajectory is 90 °. Therefore, the trajectory length is 1/4 circle, that is, (1/4) 2 π * 1=π/2. However, the actual trajectory length needs to be determined based on specific geometric constraints and folding conditions.After calculation and analysis, the trajectory length of point B is √ 2/4 π. ★ Answer: D ★

Figure 18: English Multimodal Experiment

