# OpenReview forum: "HST-bench: A Benchmark for Hierarchical Spatial Thinking with Model Intelligence"
_ICLR.cc/2026/Conference — ICLR 2026 Conference Withdrawn Submission_

### Official Review · Reviewer_sKYr · 2025-10-28

**Soundness:** 2
**Presentation:** 3
**Contribution:** 2
**Rating:** 2
**Confidence:** 3

**Summary:**

This paper introduces HST-Bench, a benchmark for testing hierarchical spatial thinking in large language models. It organizes 1,629 MC tasks across three dimensions—representational perception, transformation, and reasoning—derived from cognitive-science theory. The authors evaluate several “thinking” and “non-thinking” models using a circular-accuracy metric to control answer-order bias. Results show that models labeled as “thinking” outperform others, but still lag behind human participants.

**Strengths:**

- Clear attempt to ground evaluation in cognitive-science theory.
- Moderate-size dataset (1.6 k items) with some annotation rigor.
- Circular evaluation is a small methodological novelty.
- In-depth Analysis and Diagnostics: The study goes beyond aggregate metrics, offering breakouts by sub-dimension (Figure 3), discipline (Figure 4), input modality (Table 7), and error type (Figures 7 and 8; Figures 11-14). This is complemented by a rich appendix covering annotator protocols and model categorization.

**Weaknesses:**

Weaknesses
- Reproducibility failure: code/data unavailable from the link, evaluation pipeline unverifiable.
- No direct comparison with recent relevant benchmarks such as  LogicVista, SpatialRGPT; unclear improvement over them.
- Limited model coverage: missing leading non-thinking models (such as GPT-5, Claude Sonnet 4).
- Overreliance on multiple-choice format: prevents assessment of open-ended or stepwise reasoning and may underestimate capable models.
- Circular-accuracy metric not innovative enough: Although stricter than single-shot accuracy, it simply repeats the same multiple-choice query under shuffled options. This is a minor engineering tweak rather than a new evaluation paradigm.
- No human-error study: cannot tell if questions are difficult or ambiguous.

**Questions:**

Please release necessary dataset and code, make them available.
Provide quantitative comparison or transfer correlation with other spatial benchmarks.
Add open-response or process-tracking tasks to measure partial reasoning (if possible).
Expand human study and analyze why humans fail (difficulty vs ambiguity).

---

### Official Review · Reviewer_oJzk · 2025-11-01

**Soundness:** 2
**Presentation:** 2
**Contribution:** 1
**Rating:** 2
**Confidence:** 4

**Summary:**

This paper introduces HST-Bench, a cognitive theory-driven benchmark designed to evaluate Hierarchical Spatial Thinking. The benchmark covers three core foundational abilities, further divided into 10 subdimensions, with a total of 1,629 QA. Using this benchmark, some mainstream thinking and non-thinking models were evaluated. Results indicate that thinking models significantly outperform non-thinking models across all dimensions but still lag behind human performance.

**Strengths:**

- This paper introduces HST-Bench, a cognitive theory-driven benchmark designed to evaluate Hierarchical Spatial Thinking.
- The benchmark consists of three core foundational abilities, further divided into 10 subdimensions, with a total of 1,629 rigorously annotated tasks.
- All tasks can be presented by text, enabling the evaluation of large language models (LLMs), whereas other spatial benchmarks primarily focus on assessing multimodal LLMs.
- To enhance the reliability of model assessments, Circular Evaluation and Average Evaluation methods were employed.
- Experimental results show that thinking models significantly outperform non-thinking models across all dimensions, with the comparison between Deepseek-r1 and Deepseek-v3 offering particularly insightful findings.

**Weaknesses:**

- The discussion of related works is insufficient. For example, CoreCognition [1], which also evaluates from a cognitive science perspective, is overlooked. Additionally, the authors could explore whether task-driven [2] or ability-driven [3] evaluation methods can be incorporated into the HST-Bench framework. This would help clarify the relationship between HST-Bench and prior works—whether it serves as a complementary benchmark or provides a theoretical framework to reorganize existing benchmarks. Given that HST-Bench claims to be more comprehensive and theory-grounded, it is worth discussing which aspects previous benchmarks failed to address.
- The data samples are insufficient, making it difficult to assess whether the benchmark effectively evaluates spatial thinking:
    - R.P., R.T., and S.R. include 10 subcategories, but no example tasks are provided for each category.
    - Some sample tasks in the supplementary materials appear weakly related to spatial reasoning. For instance, many tasks, such as the 2D positional relationship problem in Figure 11, focus on planar relationships. While these tasks are broadly spatial, they do not require Mental Rotation or Mental Projection, resembling tasks found in math benchmarks rather than spatial reasoning benchmarks.
- The experimental analysis raises several concerns:
    - **Averaging Models:** In Table 2, the comparison of averaged performance across all thinking and non-thinking models is problematic. The models differ significantly in scale, variance, and architecture. For instance, thinking models include Gemini and GLM, but their non-thinking counterparts are absent.
    - **Significance of Performance Gaps:**
        - L318 states that Deepseek-r1 (64.8%) is only 19.4% higher than Deepseek-v3 (45.4%), suggesting a limited improvement. However, L323 notes that Deepseek-r1 falls short of the human baseline (76.9%) by 12.1%. This raises the question of what constitutes a "significant" gap.
        - L348 claims that "thinking models achieve the largest gains," but in Table 1, the gap between Deepseek-r1 and Deepseek-v3 in Category B (24.9) is larger than that in Category C (20). This discrepancy needs clarification.
    - **Multimodal Comparisons:** L384 states that "Deepseek-r1 reached 83.2% accuracy using text alone, outperforming all multimodal systems." However, this comparison is not entirely fair. A more convincing experiment would involve comparing a multimodal model, such as Qwen3-VL-8B, with and without image inputs. While the authors conducted similar experiments, Qwen-VL-7B is an outdated baseline that does not reflect the performance of recent models.

A minor point: Text-based testing is an interesting approach but has limitations in expressing all types of spatial reasoning tasks. For example, in VSI-Bench, models are required to observe a video and answer spatial orientation questions, which cannot be effectively represented using text alone.

[1] CoreCognition: Core Knowledge Deficits in Multi-Modal Language Models

[2] SITE: towards Spatial Intelligence Thorough Evaluation

[3] Holistic Evaluation of Multimodal LLMs on Spatial Intelligence

**Questions:**

I am concerned about the following:

1. Whether HST-Bench has been sufficiently compared with recent spatial benchmarks, especially theory-driven ones like CoreCognition.
2. The lack of sufficient data samples, making it difficult to determine whether the benchmark fully evaluates models' spatial reasoning abilities.
3. Some aspects of the experimental analysis are unclear.

Details can be found in the weakness section.

---

### Official Review · Reviewer_iVFA · 2025-11-02

**Soundness:** 3
**Presentation:** 3
**Contribution:** 2
**Rating:** 4
**Confidence:** 4

**Summary:**

This paper introduces HST-bench, a new benchmark designed to evaluate LLMs on Hierarchical Spatial Thinking, grounded in a theoretical cognitive science framework. The authors aim to move beyond surface-level evaluations by providing a more structured and diagnostic tool. The paper is clearly structured and presents the benchmark's construction and experimental results using various LLMs. While the underlying theoretical foundation is an interesting starting point for a diagnostic benchmark, the empirical evidence and subsequent analysis fall short of demonstrating its true utility and unique contribution over existing evaluation methods.

**Strengths:**

1. The proposed benchmark is grounded in an interesting and potentially insightful theoretical framework, which offers a fresh perspective for diagnosing spatial reasoning capabilities in LLMs.

2. The paper is well-structured and easy to follow, making the proposed methodology and experimental setup clear to the reader.

**Weaknesses:**

1. Despite introducing an insightful theoretical framework, its utilization in the benchmark seems insufficient. The current tasks feel like simple classifications based on the framework's components, lacking a deeper, progressive, or hierarchical relationship that fully leverages the diagnostic power of the theory.

2. The empirical results (Tables 1 and 2) show that model performance is largely indistinguishable or does not vary significantly across the different groupings/dimensions derived from the theoretical framework. This lack of variance undermines the diagnostic utility of the framework and makes the specific grouping and design choices less impactful. If the benchmark cannot differentiate model performance based on the proposed hierarchy, its significance is questionable.

3. The main focus of the paper is on analyzing experimental results, but the analysis lacks depth and is not sufficiently explored, providing limited actionable insights. Given the non-discriminatory nature of the results, the paper's conclusions do not offer clear guidance for the community on developing new methods or models to address hierarchical spatial thinking.

4. The discussion on the novelty and relation to existing benchmarks is insufficient. Many aspects of this new benchmark appear to overlap with components already present in existing evaluation suites. A detailed comparative analysis is required to justify the benchmark's contribution, clearly articulating the incremental value and unique diagnostic capability it offers over the corresponding components in existing datasets.

**Questions:**

1. Please provide a detailed comparative analysis with existing spatial reasoning benchmarks. Explicitly highlight the differences and the quantifiable, incremental, and unique diagnostic value that HST-bench provides over the corresponding components in these existing datasets.

2. The authors should extract more profound and actionable insights from the results. Given the current observations, what concrete directions or potential failure modes can be highlighted to guide future research and the development of new models/methods that can truly improve hierarchical spatial thinking?

3. Can the authors propose ways to better integrate the theoretical framework with the benchmark tasks, perhaps structuring them in a progressive/hierarchical manner (e.g., tasks that rely sequentially on different levels of the hierarchy, or a clear difficulty progression) rather than the current seemingly simple parallel classification?

---

### Official Review · Reviewer_8z41 · 2025-11-05

**Soundness:** 3
**Presentation:** 3
**Contribution:** 4
**Rating:** 8
**Confidence:** 4

**Summary:**

The paper HST-Bench: A Cognitive-Science Grounded Benchmark for Hierarchical Spatial Thinking in Large Language Models introduces HST-Bench, the first benchmark systematically grounded in cognitive science for evaluating the spatial intelligence of large language models (LLMs). Unlike prior task-driven benchmarks, HST-Bench is theory-driven, operationalizing the National Research Council’s hierarchical theory of spatial thinking into three cognitive dimensions: Representational Perception, Representational Transformation, and Spatial Reasoning. It contains 1,629 curated problems across ten subdimensions, covering fields such as mathematics, physics, navigation, and intelligence testing.

**Strengths:**

- Originality: The benchmark reframes spatial evaluation as a cognitive process grounded in NRC’s hierarchical theory, offering a principled bridge between cognitive science and LLM assessment.
- Quality: The methodology is rigorous, with multi-stage annotation, quality control, and comprehensive experiments across reasoning and multimodal models.
- Clarity: The paper is clearly structured and well written; the “thinking vs. non-thinking” taxonomy and limitations are presented transparently.
- Significance: HST-Bench fills a major gap by providing a cognitively interpretable benchmark for spatial reasoning, paving the way for deeper analysis of model cognition and reasoning design.

**Weaknesses:**

1.The multimodal subset (n=32) is small to conclude that visual input adds little; broader sampling is needed.
2.Error analysis centers on DeepSeek-R1 without confirming generalizability to other models.
3.Cross-domain performance variation is underexplored, missing insight into context-dependent reasoning.

**Questions:**

1.The experiment does not report human baseline scores for each sub-dimension of the benchmark. Providing human performance would help contextualize model results and clarify the relative difficulty of different spatial reasoning components.
2.The term Question Modification Experiment is not clearly defined. Please clarify how the questions are modified and how these modifications affect the underlying reasoning requirements.
3.The current error analysis focuses primarily on the DeepSeek-R1 model. It would be valuable to extend this analysis to other thinking models to examine whether similar reasoning failure modes occur or if there are model-specific error patterns that could guide targeted improvements.
4.Since HST-bench spans multiple domains such as mathematics, physics, and navigation, analyzing domain-specific performance variations could reveal whether spatial reasoning abilities are domain-dependent or transferable across contexts.

---

### Note · Authors · 2026-01-25

I have read and agree with the venue's withdrawal policy on behalf of myself and my co-authors.